# Cholestatic Pruritus in Children: Conventional Therapies and Beyond

**DOI:** 10.3390/biology12050756

**Published:** 2023-05-22

**Authors:** Minna Rodrigo, Xinzhong Dong, Daphne Chien, Wikrom Karnsakul

**Affiliations:** 1Division of Pediatric Gastroenterology, Hepatology, and Nutrition, Department of Pediatrics, Johns Hopkins University School of Medicine, Baltimore, MD 21205, USA; mleydor1@jh.edu; 2Department of Neuroscience, Johns Hopkins University School of Medicine, Baltimore, MD 21205, USA

**Keywords:** pediatrics, pruritus, cholestasis, liver disease, sensation

## Abstract

**Simple Summary:**

Pediatric patients with liver disease commonly experience itching, also known as pruritus. This symptom may seem to be a mere inconvenience, but can actually significantly affect the quality of life of these patients, including impairing their sleep and worsening their mental health. The cause of itching in liver disease is not fully known and is most likely caused by many factors. Unfortunately, this symptom can be incredibly difficult to treat and may ultimately require surgical interventions in certain cases that do not respond to medical therapy. Some medical therapies that are used to treat cholestatic pruritus in the adult population may be beneficial for pediatric patients suffering from this symptom. Ultimately, a better understanding of what causes itching in liver disease may provide valuable information for how to best treat this symptom.

**Abstract:**

Pruritus in the setting of cholestatic liver disease is difficult to treat and occurs in patients ranging in age from infancy to adulthood. Likely multifactorial in etiology, this symptom often involves multimodal therapy targeting several pathways and mechanisms proposed in the underlying etiology of cholestatic pruritus. Many patients in both the pediatric and adult populations continue to experience unrelenting pruritus despite maximal conventional therapy. Options are further limited in treating pediatric patients due to sparse data regarding medication safety and efficacy in younger patients. Conventional therapies for the treatment of cholestatic pruritus in children include ursodeoxycholic acid, cholestyramine, hydroxyzine, and rifampin. Certain therapies are more routinely used in the adult populations but with limited data available for use in child and adolescent patients, including opioid antagonists and selective serotonin reuptake inhibitors. Recently, ileal bile acid transport inhibitors have been shown to alleviate pruritus in many children with Alagille syndrome and progressive familial intrahepatic cholestasis and is an additional therapy available for consideration for these patients. Ultimately, surgical options such as biliary diversion or liver transplantation are considered in specific circumstances when medical therapies have been exhausted and pruritus remains debilitating. While further investigation regarding underlying etiologies and effective therapies are needed to better understand itch pathogenesis and treatment in pediatric cholestasis, current considerations beyond conventional management include the use of opioid antagonists, selective serotonin reuptake inhibitors, ileal bile acid transport inhibitors, and surgical intervention.

## 1. Introduction

Defined as the impaired secretion of bile, cholestasis commonly presents in the setting of hepatobiliary disease. This can occur due to anatomic obstruction, abnormal structures of the biliary system, infection, inflammation, or having a defective protein involved in the metabolism, transport, or excretion of the bile components. Within the pediatric population, various liver diseases may present with cholestasis with distinct characteristics of each disease as well as certain overlapping symptoms. Classic signs and symptoms associated with cholestasis include jaundice, scleral icterus, pruritus, xanthomas, steatorrhea, and failure to thrive. Cholestatic pruritus itself can be a frustrating and debilitating symptom for children with liver disease as it may be severe, unrelenting, and difficult to manage medically. Regardless of age, a notable number of patients with cholestatic liver disease experience refractory symptoms despite maximal medical management [1,2]. Within the pediatric populations, therapeutic options are even more limited due to insufficient data regarding safety and efficacy of various therapies in infants and children [3]. In children, severe pruritus is associated with functional impacts such as interference with sleep and mood disturbances [4]. Additionally, caregivers bear a notable burden of their child’s symptoms, with the severity of the child’s pruritus demonstrating strong correlation with impaired parental quality of life [5]. In certain patients with severe pruritus unable to be controlled with medications, procedural interventions such as nasobiliary and transcutaneous drainage, biliary diversion, or liver transplantation are considered to address the unrelenting itching [1,6].

Cholestatic disorders may involve intrahepatic cholestasis, which occurs due to impaired excretion of bile within the liver itself, or extrahepatic cholestasis, caused by obstruction to bile flow outside of the liver. Examples of intrahepatic cholestasis include Alagille syndrome, where patients demonstrate a paucity of bile ducts, and progressive familial intrahepatic cholestasis, caused by impairment of an enzyme crucial to bile excretion [7,8]. Extrahepatic cholestasis may be seen in anatomic anomalies such as choledochal cysts [9]. Both biliary atresia, a disease characterized by progressive obliteration of the biliary system, and primary sclerosing cholangitis (PSC), a disease involving stricturing of the biliary system within and outside of the liver, may result in both intrahepatic and extrahepatic cholestasis [10,11]. Many patients with these diagnoses experience pruritus, with 20–84% of patients with PSC reporting pruritus, and 76–80% of patients with progressive familial intrahepatic cholestasis (PFIC) experiencing pruritus [7,10]. Pruritus tends to present earlier, such as in infancy, in patients with PFIC1 and PFIC 2, whereas patients with PFIC 3 more often developed pruritus in later years [7]. In Alagille syndrome, pruritus occurs in an estimated 59–88% of patients, of which 45% are estimated to have severe pruritus [8]. The underlying cause of pruritus in cholestatic liver disease remains unknown with differing characteristics present amongst patients that experience severe itching [12]. Interestingly, patients may experience significant pruritus early on in their disease course which then subsides despite stable or even worsening cholestasis. Furthermore, even with significant cholestasis, some patients with these diseases never experience pruritus associated with liver disease [1].

## 2. Itch Pathogenesis

Cholestasis is assessed by measurement of serum conjugated bilirubin, with the measurement of other serum markers including alkaline phosphatase, gamma glutamyl transferase, and bile acid levels used to follow the disease. The correlation between these markers and cholestatic itch is poor and they are not sensitive enough to predict the occurrence of pruritus. Several studies have shown that these markers do not directly reflect the degree of pruritus experienced by the patient, with the exception of the lysophosphatidic acid (LPA)/autotaxin (ATX) axis, which partially correlates with itch sensation [13]. The complex and multifactorial nature of cholestasis may explain the poor correlation between these known serum markers and cholestatic itch. Alternatively, it is possible that there are other pruritogens that have not yet been identified, or those pruritogens interact with skin nerve endings independently of their serum levels [6]. For example, in a mouse model of cholestasis, skin bilirubin shows stronger correlation with scratching bouts than plasma bilirubin [14].

Pruritogens activate skin-innervating DRG (dorsal root ganglion) neurons, and the signal is transmitted to the second order neurons in the spinothalamic tract (STT), synapsing onto higher order neurons in the thalamus. It was once hypothesized that itch sensation is a mild form of pain and both itch and pain are encoded in the same populations of somatosensory neurons [15,16]. Based on the results of primate STT neuron recordings during the application of pruritic and noxious stimuli, in agreement with the hypothesis, pruriceptive and nociceptive signals seem to converge to some degree [17]. However, in the periphery, recent genetics and functional analyses support the notion of itch-sensing neurons as a separate entity from pain-sensing neurons, favoring “labelled line” over “intensity” theory [18,19].

Behaviorally, animals react differently upon the exposure of a pruritogen versus a pain stimulus, such as capsaicin. In mice, an intradermal injection of a pruritogen into the cheek elicits a rhythmic scratching response by only hind paws. On the other hand, capsaicin injection triggers a wiping response by front paws [20]. MRGPRA3, a murine member of Mas-related G protein-coupled receptors (MRGPRs) expressed in the DRG neurons, is activated by an anti-malaria drug, chloroquine, and mediates scratching responses in mice when given intradermally. This phenomenon recapitulates itchy side effects observed in human taking chloroquine [21]. Genetic ablation of MRGPRA3+ DRG neurons completely abolished chloroquine-induced itch while keeping capsaicin-induced pain intact [22]. Single cell transcriptome profiling of mouse DRG neurons reveals MRGPRA3 and other known itch receptors (e.g., histamine receptors, serotonin receptors, oncostatin M receptors, etc.) constitute a molecularly distinct population, named non-peptidergic (NP) neurons, different from neurons mediating other sensory modalities, such as proprioception, tactile sensation, and nociception [23]. Based on the relative expression of different itch receptors, NP neurons are subcategorized as NP1-3 with MRGPRD, MRGPRA3, and 5-hydroxytryptamine (serotonin) receptor 1F (5-HT1F) as major markers of each. Transcriptome analysis at the single cell level in non-human primates and human DRG neurons also supports the existence of pruriceptive neurons, although the molecular pattern is less well-defined compared to murine studies [24,25].

The Mas-related G protein-coupled receptor X4 (MRGPRX4) is located in the dorsal root ganglion of humans and is activated by bile acids and bilirubin. Additionally, in vivo, this receptor, when expressed in humanized mice, has been shown to cause increased itch response with exposure to bile acids and mediates itch in a mouse model of cholestasis. While therapies targeting the MRGPRX4 receptor are not currently available, this pathway may prove to be another avenue through which patients with cholestatic pruritus may be treated [26]. In mouse studies, Trans-membrane G protein-coupled receptor-5 (TGR5), also a membrane GPCR (G-protein-coupled receptors) expressed in DRG neurons, mediates bile acid induced itch [27,28]. However, from immunohistochemistry staining of human DRG tissue, TGR5 is mostly expressed in satellite glial cells but not DRG neurons [29]. So far, MRGPRX4 is the only bile acid receptor expressed on the membrane of human itch neurons.

Opioids used in pain management have been shown to have a side effect of pruritus, and endogenous opioids have been charged with contributing to cholestatic pruritus via central mechanisms. Interestingly, while both ƙ-opioids and µ-opioids inhibit pain sensation, µ-opioids may accentuate itch while ƙ-opioids can diminish this sensation. This also demonstrates the complex interaction between pain and itch pathways, even amongst the same family of signaling molecules [30].

Lysophosphatidic acid (LPA) is a signaling molecule derived from phospholipids which has been found to be elevated in patients with cholestatic pruritus. When lysophosphatidylcholine (LPC) is cleaved by autotaxin, also known as lysophospholipase D, LPA is formed. LPA plays a role in various cellular functions and pathways in addition to the proposed association with pruritus. However, studies have suggested that the LPA/ATX pathways may be distinct contributors to pathways of cholestatic pruritus in contrast to other etiologies of pruritus, such as atopic dermatitis, Hodgkin’s lymphoma, and uremia [31,32]. Elevations in serum LPA and ATX have been found to demonstrate high specificity for cholestatic itch, and ATX activity has even been found to correlate with itch intensity. The cause of elevated LPA levels in cholestatic liver disease has not been established but may be attributed to either decreased LPA clearance or increased production [6]. Studies have demonstrated increased itch response in mice injected with LPA. Neuronal mechanisms may include activation of TRPA1 (transient receptor potential ankyrin 1) and TRPV1 (transient receptor potential vanilloid 1), non-selective Ca^2+^ dependent channels which are known to contribute to both itch and pain signaling. These receptors are found in neurons of DRG, trigeminal ganglion, and nodose ganglion and in vitro have been found to be activated by LPA [31].

## 3. Management of Cholestatic Pruritus in Children

### 3.1. Ursodeoxycholic Acid

In pediatric practice, many patients are started on ursodeoxycholic acid, a synthetic bile acid, early in their disease course to improve the bile flow. While physicians may initiate this medication to alleviate cholestasis regardless of whether pruritus is present, it may also be considered as an off-label option in management of pruritus [1]. Even though this medication is technically a bile acid itself, it works to help improve hepatobiliary secretion and decrease bile toxicity [33,34,35]. As shown in Figure 1, this is prescribed in the pediatric population as weight-based dosing [1].

### 3.2. Bile Acid Binding Resins

Clinicians may also attempt to reduce serum bile acid concentrations with bile acid binding resin cholestyramine, which sequesters bile acids in a resin complex for excretion to decrease bile acid reuptake in the distal small bowel. This medication is the only one approved specifically for use of cholestatic pruritus in adults and it is often considered first line for pruritus management in the adult population [33]. As a result of decreased uptake of bile acids in the distal small bowel, this medication promotes a decreased accumulation of bile acids, thus alleviating the itch sensation in some patients. However, it rarely completely treats or controls cholestatic pruritus [1]. In conditions involving impaired secretion of bile acids into the intestines, such as biliary atresia, Alagille syndrome, or PFIC, the role of bile acid binding resins have low utility given the lack of available bile acids to bind. Furthermore, malabsorption of fats and fat-soluble vitamins may occur with this medication. Children with cholestatic liver disease are already at increased risk for fat soluble vitamin deficiency and poor weight gain due to their underlying disease [33]. Subsequently, this side effect may limit the use of this medication in the pediatric population.

### 3.3. Antihistamines

The conventional initial therapy for the symptom of pruritus in pediatric cholestatic liver disease is the use of antihistamines. Even though antihistamines such as hydroxyzine are often trialed early as pruritus therapy, the origins of cholestatic pruritus appear distinct from those seen with histaminergic itch [2,35]. Histamine is often involved in several common causes of itch, such as allergic response, urticaria, and atopic dermatitis and dosing of antihistamines is commonly based on indications for these alternative causes of itching [36]. As discussed, previous studies demonstrate alternative underlying mechanisms for cholestatic pruritus, one specific example being the family of mas-related G-protein coupled receptors, including MRGPRX1 and MRGPRX4 as itch receptors expressed in dorsal root ganglions of humans mediating non-histaminergic itch [6,37]. Even so, histamines are often trialed with limited effectiveness and the commonly experienced side effect of drowsiness [33,34].

### 3.4. Rifampin

Rifampin is often a second line therapy for treatment of cholestatic pruritus in children and produces notable improvement in pruritus in many patients. However, rifampin’s mechanism of action in the treatment of cholestatic pruritus has not been definitively determined. Some proposed mechanisms of action of this therapy include inhibiting transcription of autotaxin, thus mediating the LPA/ATX pathway [31,33]. Additionally, rifampin may play a role in the activation of the nuclear pregnane X receptor which enhances enzymatic reactions that make bile acids more hydrophilic and less toxic. This is thought to allow for increased elimination of bile and bilirubin [1,38]. Common dosing in pediatric patients is shown in Figure 1. When using rifampin, side effects may include nausea, decreased appetite, risk of hepatitis [1,34,35].

Rifampin has been noted in the literature to have potential therapeutic use specifically in cases of extrahepatic cholestasis. Alternative agents, such as opioid antagonists, have been shown in some studies to have little to no effect at treating pruritus in the setting of extrahepatic cholestasis, whereas rifampin was effective in these cases [39]. Other sources have noted the utility of rifampin in treating cholestatic itch in cases of primary biliary cholangitis (PBC) and intrahepatic cholestasis of pregnancy [40,41]. While these conditions are more commonly seen in the adult population, the effective use of rifampin in these disorders demonstrates the potential of this medication for a spectrum of conditions demonstrating cholestatic itch.

### 3.5. Opioid Antagonists

Patients with cholestatic pruritus have been known to respond to opioid antagonists such as naltrexone with relief of their pruritus [42]. Additionally, patients with cholestatic pruritus have even demonstrated symptoms of opioid withdrawal such as tachycardia, hypertension, abdominal pain, and piloerection when initiating opioid antagonist therapy, suggesting increased opioidergic tone at baseline in these patients [6,42]. The mechanisms for this heightened opioid pathway are unknown, but with consideration of endogenous opioids being produced in the liver in the setting of cholestasis [42]. Opioid antagonists can be administered either by IV (intravenous injection) or PO (per os) for treatment of cholestatic pruritus. However, one concern with this therapy is the development of tolerance to this medication due to continued exposure to opioid antagonists, causing decreasing efficacy of opioid antagonist therapy [6]. Sparse data regarding the use of opioid antagonists in the management of cholestatic pruritus in children is currently available in the literature and this medication has not been approved by the United States Food and Drug Administration (FDA) in patients under 18 years of age [34,35]. However, naltrexone has been used to treat cholestatic pruritus in patients aged as young as 17 months, per case report [43]. Weight based dosing is typically used in the pediatric population, as shown in Figure 1 [1,34,35].

While some consider opioid antagonists to have little utility in treating pruritus in cases of extrahepatic cholestasis, with a potential for better therapeutic effect in cases of intrahepatic cholestasis, the multifactorial nature of pruritus, with variation even between individuals with the same disease, suggests a benefit to following a step-wise approach with consideration of opioid antagonists in those that do not fully respond to rifampin, regardless of the extrahepatic or intrahepatic nature of the cholestasis [39,44].

### 3.6. Selective Serotonin Reuptake Inhibitors

While levels of serotonin are not consistently correlated with presence or severity of cholestatic pruritus, the use of selective serotonin reuptake inhibitors has been seen to alleviate pruritus in cases refractory to other treatment regimens and is used as a fourth line in treatment of cholestatic pruritus in adult patients [6,34]. One study investigated the use of sertraline for treatment of pruritus in pediatric patients with Alagille syndrome or PFIC, with improved pruritus noted in 14 of the 20 patients treated with the medication. In this study, 10 of the 20 patients were classified as responders based on criteria determined prior to commencement of the study, including outcomes of improved pruritus with the additional endpoints of either improved skin scratching score or improved sleep score. Three patients experienced adverse events leading to discontinuation of the medication, including agitation, skin reaction, and vomiting. In this study, the youngest patient was 1.8 years old. While the study does not note which patients required discontinuation of the medication, it does note that adverse events were reversible [3]. Previously used weight-based dosing is shown in Figure 1 [1,3].

### 3.7. Ileal Bile Acid Transport (IBAT) Inhibitors

New pharmacologic interventions targeting enterohepatic bile acid circulation have been developed and are currently used in the management of Alagille syndrome and PFIC specifically, as shown in Figure 1. Maralixibat has recently been approved by the United States FDA as an IBAT inhibitor used in Alagille syndrome, aimed to decrease enterohepatic bile acid circulation, thus decreasing bile acid stores and alleviating burdens of cholestasis including a potential therapeutic role for cholestatic pruritus. Similarly, odevixibat has been FDA approved for management of PFIC [34,45]. Maralixibat was found to demonstrate improved serum bile acid levels and statistically significant reduction in pruritus in patients with Alagille syndrome [46]. Furthermore, clinical trials have shown improved growth for patients with Alagille syndrome and PFIC on maralixibat [46,47]. In clinical trials, odevixibat has been shown to decrease bile acid levels and pruritus severity, in addition to improving sleep quality in patients with Alagille syndrome, PFIC, and biliary atresia. While IBAT inhibitors may have a broader role in the management of Alagille syndrome and PFIC, trials have demonstrated benefits specifically regarding improved pruritus in these populations [34,45,46,47].

### 3.8. Surgical Management

Patients with certain pediatric cholestatic liver diseases, specifically including PFIC and Alagille syndrome, who are refractory to medical management of pruritus may benefit from surgical intervention to treat their itching. Biliary diversion functions to reduce enterohepatic circulation, thus decreasing retention of certain substances including bile acids and bile salts which may contribute to pruritus [1,48]. Prior to surgery, some patients first trial nasobiliary drainage to gauge the degree of improvement that surgical intervention may provide. Nasobiliary drainage involves endoscopic placement of a nasobiliary tube which then drains substances from the biliary system, decreasing the amount of potential pruritogens in the body. Partial external biliary diversion involves the creation of a conduit allowing for a notable portion of the volume of bile flowing from the liver to be redirected externally, thus minimizing enterohepatic circulation of pruritogens. Partial internal biliary diversion serves as an alternative surgical option and classically involves use of an isolated portion of jejunum as an internal conduit between the gallbladder and colon, thus avoiding the distal small bowel where reuptake of bile acids occurs [48].

Surgical interventions clearly involve risks, complications, and limitations. Partial external biliary diversion involves the creation of an external stoma which requires care and attention. Stoma output may at times be high in volume and result in electrolyte abnormalities and dehydration. Other complications may involve para stomal hernia, cholangitis, and small bowel obstruction. Partial internal diversion has been trialed as a surgical technique relatively recently compared to external diversion, resulting in less known data regarding risks and potential complications. However, similar to partial external diversion, one known risk of partial internal diversion includes that of adhesive small bowel obstruction. More commonly, patients experience choleretic diarrhea due to the high load of bile salts diverted to the colon. Theoretical risks of increased colon cancer due to significant colonic exposure to bile acids as well as the hypothesized increased risk of cholangitis have not yet been demonstrated in the literature [48].

Ultimately, liver transplantation is considered in some pediatric patients with cholestatic liver disease who suffer from pruritus. Transplantation may ultimately be indicated regardless of pruritus severity when liver disease progresses to cirrhosis. In certain cases where significant progression of liver disease is predicted, transplant in the setting of severe pruritus may be considered to definitively treat the pruritus without first performing other surgical interventions such as biliary diversion [6,48].

**Figure 1 biology-12-00756-f001:**
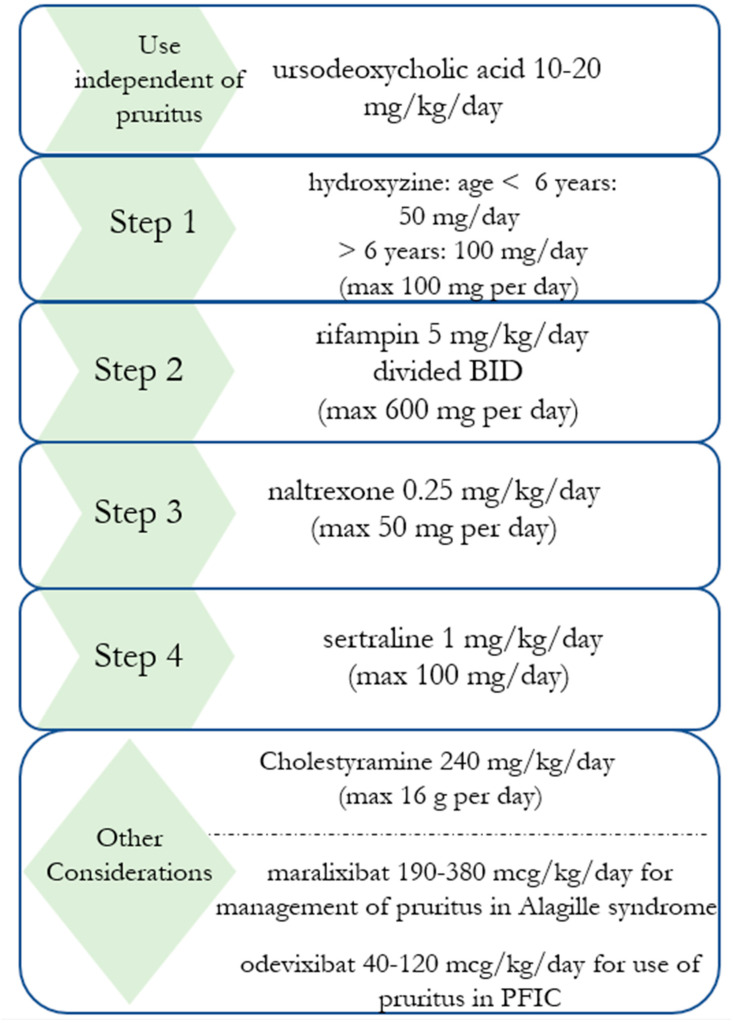
Figure 1 demonstrates suggested treatment considerations for management of cholestatic pruritus in children. Ursodeoxycholic acid is used in various forms of cholestatic liver disease and not solely used as pruritus treatment, but may have a role in alleviating pruritus in these patients [1]. Step 1: antihistamines may have limited effectiveness in cholestatic itch but are often tried initially due to tolerability. Side effects may include drowsiness [35,36]. Step 2: rifampin: may increase to 10 mg/kg/day divided BID if not responsive to initial dose [1]. Side effects may include nausea, decreased appetite, risk of hepatitis [34,35]. Step 3: naltrexone: side effects may include nausea, abdominal pain, headache, dizziness; not FDA approved for age <18 [1,34,35]. Step 4: sertraline: side effects may include agitation, skin reactions, and vomiting [3,34]. Other considerations: Cholestyramine may have a role in managing cholestatic itch, but use in the pediatric population may be limited due to side effects of malabsorption of fats and vitamins [1,34,35]. Maralixibat for use in patients with Alagille syndrome approved by United States FDA in age 12 months and older; odevixibat for use in patients with PFIC age 3 months and older as approved by United States FDA [34,45,46,47]. Theoretical risk of fat-soluble vitamin deficiency with use of IBAT inhibitors due to decreased bile acid pools, but not seen in long term study of maralixibat [47]. Close monitoring of fat-soluble vitamin levels recommended with IBAT inhibitors.

## 4. Discussion

Cholestatic pruritus remains challenging to treat in children predominantly because the underlying cause of this itching remains unknown and is likely multifactorial [1]. This symptom may be considered benign when mild, but in many cases significantly affects patients’ quality of life through a substantial negative impact on sleep and mood [4].

Even though pruritus of varying etiologies may present clinically with a similar sensation, these different forms of pruritus are distinct. Systemic pruritus, which differs from pruritus attributed to focal dermatologic conditions, may occur in a variety of disorders including cholestasis, chronic kidney disease, hematologic disease, and malignancy, amongst others. In these conditions, there is at baseline no skin disease or dysfunction, but with continued scratching, secondary skin excoriation, hyperpigmentation, and scarring may occur. Some therapies are used across many disorders causing systemic pruritus. However, different treatment modalities produce different degrees of effectiveness amongst various disorders of systemic itch, due to different underlying mechanisms at work [12]. For example, the origins of cholestatic itch are distinct from those seen with histaminergic itch, such as allergic response, urticaria, and atopic dermatitis. Histamine release by mast cell degranulation causes “wheel and flare” erythematous reactions in the skin, which are rarely seen in patients with cholestatic pruritus. Furthermore, plasma levels of histamine do not correlate with itch intensity in patients with various cholestatic diseases [13]. A study by O’Keeffe et al. has shown that there is no alteration in cutaneous mast cell density in patients with chronic liver disease with or without pruritus or healthy controls [49]. There are likely both central and peripheral sensory components which contribute to cholestatic pruritus. It seems that pruritogens synthesized in the liver and present within bile overflow into the hepatic sinusoids and then into systemic circulation, depositing in tissues such as the skin, which thus contributes to sensation of pruritus [6]. When an anatomic obstruction or blockage occurs, such as a stone or stricture, patients may experience pruritus that is then relieved upon correction of the obstructing factor. Additionally, when patients with cholestatic liver disease experience worsening of their disease and even progression to cirrhosis and liver failure, severe pruritus often abates even in the presence of worsening cholestasis. These principles suggest that buildup of pruritogens produced in the liver contributes to symptoms of pruritus but the liver must have some degree of function for the pruritus to occur [2].

Therapies used in cholestatic liver disease, including ursodeoxycholic acid and cholestyramine, are now well understood in their mechanism of action but may have limited utility in satisfactory treatment of pruritus in this setting. Additionally, while cholestyramine is approved for the treatment of cholestatic pruritus in adults, concerns regarding fat and vitamin malabsorption due to usage of bile acid binding resins may limit the benefits of this medication and it should be initiated thoughtfully [34]. Management of pruritus in pediatric patients with cholestatic liver disease should be conducted in a step-wise fashion, and polypharmacy is often required to address the multifactorial etiology of this symptom. A notable number of pediatric patients who experience pruritus in the setting of cholestasis remain refractory to conventional pruritus management, prompting consideration of novel therapies including the opioid antagonist naltrexone, and the selective serotonin reuptake inhibitor sertraline [1]. These therapies have become mainstays in the step-wise management of pruritus in adult patients with cholestatic liver disease, but with minimal data at present pertaining specifically to pediatric patients [34]. These therapies may be considered but with caution, and close monitoring upon initiation of these medications for pruritus management is advised. New therapeutic options in the pediatric population include ileal bile acid transport inhibitors; maralixibat and odevixibat have recently been FDA approved for use in children with Alagille syndrome and PFIC, respectively, and have thus far demonstrated promising data regarding improvement in pruritus, amongst other end points. Surgery may be required in specific circumstances when patients are refractory to all medical management, but does involve notable risks and limitations [48]. Ultimately, liver transplantation is considered in select patients suffering from pruritus in the setting of significantly impaired quality of life and/or expected progression of liver disease [13].

Further exploration of itch pathogenesis will hopefully provide maximally effective therapeutic options. The level of bile acids in the skin of cholestatic patients has been studied [50,51]. While there is no strong correlation of skin bile acids and itch severity in cholestatic patients based on current studies, it is worth noting that bile acids are composed of multiple species (both conjugated and unconjugated) and may activate receptors, such as MRGPRX4, at varying potencies [51]. Fractionated bile acids can be obtained to differentiate between different classes of bile acids. Instead of using total bile acid levels in the skin as a surrogate for cholestatic pruritus, it may be more informative to focus on the specific bile acids that can activate receptors potently. For example, the discovery of MRGPRX4 as a receptor on the dorsal root ganglion of humans shown to be activated by bile acids may be relevant to itch pathogenesis, and additionally a target for therapeutic management of cholestatic pruritus. While serum bile acids have not been shown to be a direct marker for presence or severity of pruritus, the presence of bile acids in the skin may contribute to one facet of this pathway and further investigation is warranted [2,19,26,29].

Additional investigation targeting disease-specific pathogenesis may also provide promising therapies in some cases. Sodium-4-phenylbuturate (NaPB), a medication used in the management of urea cycle disorders, has been shown to have beneficial effects in patients with PFIC1 and PFIC2 due to increased hepatocanilicular expression of bile salt export pump (BSEP), the protein affected in these conditions [52]. Specifically, the use of NaPB in PFIC2 patients with impaired BSEP expression resulted in improved pruritus and liver histology [52,53,54]. Another study demonstrated resolution in pruritus for patients with PFIC1 initiated on NaPB but without improved histology or biochemical liver markers [55].

Bezafibrate, a broad peroxisome proliferator-activated receptor (PPAR) agonist has been studied in adult patients with PSC and PBC for treatment of cholestatic pruritus. While both conditions are progressive fibrosing cholangiopathies, PBC results in intrahepatic cholestasis caused by interlobular bile ductule destruction, whereas PSC occurs due to stricturing of intrahepatic bile ducts, extrahepatic bile ducts, or a combination of the two. PPAR functions as a transcription factor which functions to activate both fatty acid catabolism and the inflammatory response. Additionally, these agonists may also decrease bile acid synthesis in the liver, which may in part relieve cholestasis [56]. Studies in the adult population demonstrate a notable antipruritic effect in many patients with either PSC or PBC. A double-blind randomized placebo-controlled trial which involved patients with either PSC or PBC demonstrated that 55% of PBC patients who received bezafibrate experienced reduction in pruritus compared to 13% of patients receiving a placebo, while 41% of PSC patients receiving bezafibrate experienced reduction in pruritus compared to 11% of patients receiving a placebo [56]. Additionally, serum markers such as alkaline phosphatase were also shown to decrease with the use of bezafibrate [56,57]. This did not correlate with a decrease in ATX activity or serum bile acid levels [57]. However, it should be noted that the use of bezafibrate in these conditions is still under investigation in the adult population specifically to better understand safety for use in patients with liver disease, and thus has not yet been considered for standard use in the pediatric population [56].

## 5. Conclusions

Cholestatic pruritus remains difficult to treat with a likely multifactorial etiology. Conventional therapeutic options for cholestatic pruritus in children include bile acid binding resins, ursodeoxycholic acid, antihistamines, and rifampin. However, many patients’ pruritus remains refractory to these therapies, and novel therapies including opioid antagonists and selective serotonin reuptake inhibitors may be considered. In certain circumstances, the incessant itching requires surgical intervention, such as biliary diversion, or even liver transplantation to definitively treat this symptom [1,6]. Despite significant advances in the care of pediatric patients with cholestatic liver disease, much remains to be discovered regarding the treatment of cholestatic pruritus.

## Data Availability

No new data were created or analyzed in this study. Data sharing is not applicable to this article.

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
