# Peer review of "Cholestatic Pruritus in Children: Conventional Therapies and Beyond"

_biology, 2023, doi:10.3390/biology12050756_

Round 1

Reviewer 1 Report

This is a great mini review on pruritus in adolescent cholestasis.

Two main concerns:

1. Title is misleading from the content of the review

2. More resources need to be cited. It would be beneficial to expand as much as possible since childhood pruritus and treatment of cholestasis is not well know and this review will help readers understand the gap in knowledge for researchers and clinicians alike.

Reviewer 2 Report

I read the manuscript “A Novel Concept for Management of Cholestatic Pruritus in Children” of Rodrigo et al. with great interest. The review enumerates clinical challenges associated with cholestatic pruritus in children and adolescents. Moreover, previously proposed potential mediators implicated in pruritus genesis in cholestasis are summarized thoroughly. Furthermore, the manuscript gives a step-up-therapy recommendation in the context of cholestatic pruritus in pediatric patients. This review was written by researchers specialized in pediatric hepatology and neuroscience.

Major comments:

1)     It is unclear what the novelty of this manuscript in terms management of cholestatic pruritus in children is. Especially differences in contrast to preexisting recommendations should be highlighted. Why do the researchers believe that their approach is better than others? What is this assumption based on? The manuscript in its current form does hardly support the claim of a “novel concept” as stated in the title.

2)     The abstract is rather vague and does not provide enough specifics pertaining the novelty of their approach.

3)     The introduction is quite extensive and should be shortened. As the discussion is relatively short some parts of the introduction might fit in the discussion.

4)     The authors should distinguish between intrahepatic and extrahepatic cholestasis. It was suggested that patients with extrahepatic cholestasis might benefit from the addition of or switch to rifampin, whereas patients with intrahepatic cholestasis might benefit from an opioid antagonist or phenobarbital. (Kriegermaier et al. Treatment of cholestatic pruritus in children, 2007 DOI: 10.2146/ajhp060453) Can the authors comment on that?

5)     Pg. 3: While there is no strong correlation of skin bile acids and itch severity, it is worth noting that bile acids are composed of multiple species (both conjugated and unconjugated) and activate receptors at varying potencies. Instead of using total bile acid levels in the skin as a surrogate for cholestatic pruritus, it may be more informative to focus on the specific bile acids that can activate receptor potently.  ïƒ  It should be clarified what receptors the authors are actually referring to. FXR? TGR5?

6)     The authors summarize the neuronal mechanisms of bile acid or bilirubin-induced itch. On the other hand they rightfully state that those markers do not directly reflect the degree of pruritus experienced by the patient. Hence, it would be more interesting to elaborate on the neuronal etiology of other potential pruritogens (e.g. LPA).

7)     The authors mention MRGPRX1 as an itch receptor while they only describe MRGPRX4 in itch pathogenesis. How do those two receptors differ? Are they redundant receptors?

8)     The term primary sclerosing cholangitis (PSC) is firstly mentioned on page 2, paragraph 3. However, the term PSC is used predominantly in adults. In pediatrics, sclerosing cholangitis typically presents with additional features like those of autoimmune hepatitis (AIH). In a pediatric manuscript the term PSC/AIH-Overlap or ASC seems to be more appropriate.

9)     Page 2, paragraph 3: It should be mentioned that pruritus severity in PFIC depends on the subtype.

10)   Page 4, point 3.2.: It should be highlighted that in cholestatic states in which the secretion of bile acids in bile as well as in intestine is low (e.g., biliary atresia, Alagille syndrome or PFIC) cholestyramine or comparable drugs have low efficacy.

11)   4-phenylbutyrate has been examined in children suffering from PFIC2. An association with re-expression of BSEP and improvement of pruritus has been described in those patients. Moreover, bezafibrate, a broad peroxisome proliferator-activated receptor agonist, has been shown to reduce moderate and severe pruritus in adult PSC and PBC patients. Those therapeutic options should be mentioned in the manuscript as well.

Minor comments:

1)     Page 7: The paragraph “Other considerations: Ileal bile acid transport inhibitors maralixibat and odevixibat have been FDA approved for use of pruritus in Alagille syndrome and PFIC respectively. Notable side effects include diarrhea and abdominal pain” has been discussed on page 5/6. In my opinion it should be deleted on page 7.

2)     Data presentation: Unfortunately, the figure is pixelated. Therefore, I would recommend a higher resolution.

Round 2

Reviewer 2 Report

All points raised have been adequately answered in this revised version. I recommend publication.